# Effect of Puncture Vine on Growth Performance, Carcass and Meat Traits, Metabolic and Immunological Blood Indicators, and Selected Cecal Microbiota in Broiler Chickens

**DOI:** 10.3390/ani13233708

**Published:** 2023-11-29

**Authors:** Hani H. Al-Baadani, Abdulrahman S. Alharthi, Saleh Al-Ghamdi, Ibrahim A. Alhidary

**Affiliations:** 1Animal Production Department, College of Food and Agriculture Science, King Saud University, P.O. Box 2460, Riyadh 11451, Saudi Arabia; abalharthi@ksu.edu.sa; 2Agricultural Engineering Department, College of Food and Agriculture Sciences, King Saud University, P.O. Box 2460, Riyadh 11451, Saudi Arabia; sasaleh@ksu.edu.sa

**Keywords:** *Gallus domesticus*, *Tribulus terrestris*, performance, carcass, microbiota

## Abstract

**Simple Summary:**

Puncture vine has recently been used as a natural alternative to antibiotics in the broiler industry, but studies are very limited. However, the goal of this research was to determine the practical and safe use of puncture vine in broiler chickens by assessing its efficacy on the performance, carcass and meat characteristics, selected metabolic and immunological blood indicators, some microbiota, and short-chain fatty acids. The current study will help workers and researchers determine the most effective or least harmful amount of puncture vine for broiler chicken performance and health. In conclusion, the dosage of 0.08% has a safe effect on the performance, carcass and meat characteristics, and health of broiler chickens. Further studies on toxicity on the different organs of broiler chickens are needed.

**Abstract:**

This study investigated the effects of puncture vine (*Tribulus Terrestris*) addition on the performance, carcass and meat characteristics, selected metabolic and immunological blood indicators, some microbiota, and short-chain fatty acids. A total of 252 1-day-old broilers were distributed to three treatments with 12 cages as replicates per treatment (T1 = 0.0%, T2 = 0.08%, and T3 = 0.16% puncture vine). Performance parameters and metabolic and immunological serum indicators were measured in each feeding phase, while carcass characteristics, meat quality, cecal microflora, and short-chain fatty acids were measured at 35 days. Results showed that live weight, weight gain, production efficiency, and meat component color were lower in initial and ultimate at T3, while the percentages of the legs and gizzard were higher at T2 than T1. The relative weight of cooking loss was higher in T2 and T3, but the myofibril fragmentation index was lower than T1. Total protein and globulin were higher in T2 and T3 (14 days old), and the glucose level was lower in T2 (35 days old) than at T1. Interleukins (*IL-4*, *IL-6*, and *IL-10*) and tumor necrosis factor-α (*TNF-α*) were higher in T2 than T1 and T3 (35 days old). Puncture vine has antimicrobial activity against *Escherichia coli* and *Salmonella* spp., while *Lactobacillus* spp. was higher in T2. The total short-chain fatty acid content was higher in chickens fed puncture vine. These results indicate that the use of puncture vine powder as a natural alternative at a dosage of 0.08% has a safe effect on the performance, carcass and meat characteristics, and health of broilers.

## 1. Introduction

Broiler chickens have attracted increasing attention worldwide in recent years due to their tender meat, nutritional quality, and low cost. They are considered the most important source of protein for human growth and development compared to the meat of other animals [1]. However, improving the production and health of broiler chickens has become of interest in many recent studies to achieve sufficient food security [2]. Therefore, antibiotics continue to be used in the poultry industry at sub-therapeutic doses as growth promoters to improve health, growth rate, and feed efficiency [3]. Nevertheless, the increasing prevalence of antibiotic-resistant bacteria and their side effects on human and animal health has led to a renewed interest in medicinal plants as natural alternatives to antibiotics [4]. Developing countries have a great potential of medicinal and aromatic plants [5], and previous studies have demonstrated the efficacy of using medicinal plants and their extracts in poultry nutrition [6].

Puncture vine (*Tribulus terrestris*), a plant from the Zygophyllaceae family, is commonly known as puncture vine. It has been used for various health problems affecting the liver, kidneys, cardiovascular system, and immune system. The fruit of puncture vine is widely distributed in the southern part of Europe, Africa, and the western part of Asia [7]. Puncture vine fruit is widely used in traditional Indian and Chinese medicine as an anthelmintic, antifungal, and antibiotic in humans [8]. In addition, puncture vine and its extracts are an alternative medicine for many diseases, including colic, hypertension, hypercholesterolemia, diabetes, and diseases of the cardiovascular and respiratory systems, and are commonly used as a diuretic [9,10].

The phytotherapeutic potential of puncture vine is contained in its leaves, roots, seeds, and fruits [11]. Its active constituents include nitrate, terrestroside, dioscin, alkaloids, flavonoids, saponins, tannins, resin, kaempferol, gitogenin, harmane, harmine, and chlorogenin [12,13]. Puncture vine is used in various industries, including cosmetic and pharmaceutical industries based on its saponin fraction [14] and as a food additive in Europe and the USA [15]. The saponins are considered useful for research to rule out alternatives for the control of poultry diseases [16].

A study by Şahin [17] showed that puncture vine powder (0.8 g per kg of basal diet) can be used in broiler diet as an alternative to antibiotics. Several studies also showed the positive effects of puncture vine extracts on the productivity and health of broilers [18,19]. Yazdi et al. [20] indicated that puncture vine powder had a positive effect on the production performance and immune responses of broiler chickens. On the other hand, puncture vine extracts at dosages of 0.06 and 0.12 g/kg diet did not change the growth performance and carcass composition in broiler chickens [21]. In addition, the use of puncture vine as a powder at dosages of 1 and 2 g/kg diet also had no effect on the production performance of broilers [22]. The puncture vine extract was found to have dose-dependent immunoregulatory properties compared to the control group [23]. Many studies have found that the powder and extract of puncture vine have an antimicrobial effect [24,25].

Despite previous studies that demonstrated the effect of the bioactive components of puncture vine on broiler chickens, the underlying mechanisms of action and the optimal dosage application have not yet been clarified. The aim of this study was to provide evidence for the future use of puncture vine powder as a natural alternative to improve the performance, meat quality, and health indicators of broiler chickens based on the previously investigated bioactive compounds of this plant. Therefore, we investigated the use of puncture vine powder as a whole plant in broiler chickens to assess its effects on growth performance, carcass and meat characteristics, metabolic and immunological blood indicators, and selected cecal microbiota and short-chain fatty acids.

## 2. Materials and Methods

### 2.1. Ethical Approval and Preparation of Puncture Vine Powder

This study was approved by the Ethics Committee for Scientific Research of King Saud University (KSU-SE-21-47). Puncture vine (*Tribulus terrestris*) was collected from the valleys of the Ibb city in the Republic of Yemen. The plant was identified and accessioned in the herbarium of the College of Science-KSU (No. 24519). It was then dried at 27 °C to a constant weight and ground into powder at the Department of Animal Production-Food and Agricultural Sciences College. The bioactive ingredients of puncture vine were analyzed by gas chromatography–mass spectrometry (GC–MS, Agilent Technologies, Palo Alto, CA, USA) according to the method described by Cheng et al. [26].

### 2.2. Birds, Feeding, and Husbandry

In this study, 252 commercial broiler chicks (Ross 308) were used, weighed individually at one day of age, and randomly divided equally to three dietary treatments with 12 cages as replicates per treatment (seven chicks per cage). The diet treatments were supplemented as follows: T1 = 0.0%, T2 = 0.08%, and T3 = 0.16% puncture vine (0.0, 0.8, and 1.6 g puncture vine per kg of basal diet, respectively) for 35 days. The basal diet was formulated in two phases from 0 to 14 days old (starter) and 15 to 35 days old (grower) in the mashed form based on soybean meal and yellow corn according to the commercial recommendations used in Saudi Arabia (Table 1). Feed and water were given ad libitum during the experimental period. All birds were housed in environmentally controlled battery cages where the temperature was 35 °C on the first day, which was gradually reduced (2 °C/3 days) to 22 °C on the 21st day. Vaccination against NDV, IBV, and IBDV was administered to all birds at the appropriate age by nasal administration according to the manufacturers’ recommendations (Fort Dodge Animal Health, Fort Dodge, IA, USA).

### 2.3. Growth Performance Evaluation

All birds were weighed individually to determine body weight at hatching (1 day) and at the end of each feeding phase (starter and grower). Body weight gain (BWG = body weight at the end of each feeding phase − initial body weight) and total feed intake (TFI = feed submitted − feed remaining) were recorded to determine feed conversion ratio (FCR = feed intake/weight gain) according to Jimoh et al. [27]. Meanwhile, the production efficiency factor (PEF = (livability × live weight (kg))/(age in days × FCR) × 100) was calculated [28].

### 2.4. The Slaughter Process, Carcasses, and Meat Traits

At the end of the grower feeding phase (35 days old), two birds per cage (24 birds per dietary treatment, 72 total) were randomly selected for slaughter using standard commercial procedures at the slaughterhouse after depriving all birds of feed with access to water for 12 h in accordance with animal welfare regulations, and live weight was recorded before slaughter. The live weight and carcass weight were used to calculate the percentage of carcass dressing (carcass weight/live weight × 100). Breast, legs, gizzard, liver, heart, pancreas, abdominal fat, thymus, bursa, and spleen were weighed separately and calculated based on live weight (body parts and organ weight/live weight) × 100 according to Bai et al. [29].

A digital pH meter (Hanna Instruments, Padova, Italy) and a digital colorimeter (Konica Minolta, Tokyo, Japan) were used to measure pH, and the color values (lightness, redness, and yellowing) of the breast muscle were recorded twice at 15 min and 24 h after slaughter (initial and ultimate) using a digital colorimeter (Konica Minolta, Tokyo, Japan) according to the procedures described by Hussein et al. [30]. The filter paper press method was used to measure the water-holding capacity. In this method, the filter paper was pressed with a force of 12 kg for 5 min. The meat was weighed again and the difference between the weights was expressed as a percentage [31].

Approximately 4 g of the breast meat was homogenized in 40 mL of cold buffer (4 °C) and then washed several times to determine the myofibrillar fragmentation index using a spectrophotometer analyzer (JENWAY, 6705, Stone, Staffs, UK) and absorbing 1 mL of the solution at 540 nm [32]. Then, 100 g of the breast meat sample was heated in an electric oven to a temperature of 70 °C to calculate the cooking loss. The difference between the initial weight and the cooked weight divided by the initial weight was used as the basis for calculating the percentage of cooking loss [31].

Shear force was determined by cutting the cooked meat sample into longitudinal muscle fiber pieces (2 cm^2^) and measuring the force required to cut through the sample using a TA-HD Texture Analyzer (Stable Micro Systems Ltd., Godalming, UK) with a Warner–Bratzler attachment [33]. The texture profile analysis (TPA), including hardness, springiness, cohesiveness, and chewiness, was determined using the TA-HD Texture Analyzer (Stable Micro Systems Ltd., Godalming, UK) outfitted with a Warner–Bratzler attachment and a compression plate for each test [34].

### 2.5. Sample Collection and Analysis of Blood Indices

Blood samples (two birds per cage, 24 birds per dietary treatment) were randomly collected at the end of each feeding phase (14 and 35 days) into tubes containing ethylene diaminetetraacetic acid as an anticoagulant. The serum was separated by centrifugation at 1500× *g* for 20 min and frozen at −20 °C until analysis.

Serum biochemical concentrations (metabolic blood indicators) were measured for total protein, albumin, glucose, and total cholesterol. According to the manufacturer’s instructions, these biochemical parameters were analyzed by spectrophotometric (RANDOX, UK) using reagent kits (Randox, London, UK). Serum globulin was determined by subtracting the albumin concentration from total protein, as previously described by Olasehinde and Aderemi et al. [35].

The concentrations of tumor necrosis factor-α (*TNF-α*) and interleukins (*IL-4*, *IL-6*, and *IL-10*) were measured in serum using enzyme-linked immunosorbent assay kits (C Bioassay Technology Laboratory, Zhejiang, China) and a microplate reader (MR-96A; Mindray Bio-Medical Electronics Co., Ltd., Shenzhen, China), as previously described by Xie et al. [32].

### 2.6. Examination of the Selected Cecal Microbiota

At the end of the grower feeding phase (35 days old), the cecal contents of two birds per cage (24 birds per dietary treatment) were sampled after slaughter to measure the concentration of microorganisms (pathogenic and non-pathogenic bacteria) according to the published method [36]. Briefly, nearly 1 g of the cecal content was serially diluted in 9 mL of buffered peptone water (1:10) to the desired dilution and then cultured on selective agar media for the bacterial species studied. Selective agar media were used for enumeration of bacterial targets such as *Lactobacillus* spp. on MRS agar (Himedia, Mumbai, India) at 37 °C with 5% CO_2_ for 24 h, while total aerobes, *Salmonella* spp., and *Escherichia coli* were cultured on nutrient agar, SS agar, and McConkey agar (Himedia, Mumbai, India) at 37 °C for 24 h, respectively. Colonies were counted using a colony counter and the results were expressed as log10 colony-forming units per gram (log^10^ colony-forming units/g).

### 2.7. Analysis of Cecal Short-Chain Fatty Acids

At the end of the grower feeding phase (35 days old), the cecal contents of two birds per cage were sampled to determine the concentration of short-chain fatty acids (acetic acid, propionic acid, and butyric acid) using gas chromatography–mass spectrometry (Agilent Technologies, Palo Alto, CA, USA) according to the published method [37]. The concentration of acetic acid, propionic acid, and butyric acid was expressed as a percentage per 100 g of total short-chain fatty acids.

### 2.8. Statistical Analysis

All results of this study, including performance, carcass characteristics, meat quality, blood indicators, microbiota, and short-chain fatty acids, were analyzed using the general linear model (GLM) of the Statistical Analysis System software 9.3 [38] by one-way ANOVA for a completely randomized design (CRD) in an environmentally controlled chamber. The following statistical model was used:

Observed values (Y*_ij_*) = general mean (μ) + the effect of dietary treatment (T*_i_*) + the random error (e*_ij_*).

Normality of data was tested (Shapiro–Wilk test) before statistical analysis was performed. Significant differences between means were analyzed by Duncan’s multiple range tests, with statistical significance based on *p* < 0.05. The standard error of the mean (±SEM) was reported for all means of each parameter.

## 3. Results

### 3.1. Bioactive Compounds Identified of Puncture Vine

The results of the bioactive compounds identified in the methanolic extract of puncture vine by gas chromatography–mass spectrometry analysis is reported (Table 2). The puncture vine extract consisted mainly of phenolic and flavonoid compounds, with 5-octadecene and hexadecanoic acid being the most abundant residue and accounting for 21.35 and 16.35% of the bioactive compounds in puncture vine, respectively.

### 3.2. Performance Measurements

The results obtained by the addition of puncture vine on the performance of broiler chickens during the feeding phases are presented in Table 3. In all diet treatments, body weight (BW), body weight gain (BWG), total feed intake (TFI), feed conversion ratio (FCR), and production efficiency factor (PEF) were not affected in the starter phase (1–14 days old) (*p* > 0.05). However, in the grower phase (15–35 days old), BW and BWG were lower (*p* = 0.0003 and *p* = 0.002, respectively) in chickens receiving T3 (0.16% puncture vine) compared to T2 (0.08% puncture vine) and the control group (T1), while T2 was not significantly different compared to T1. PEF was higher (*p* = 0.004) in chickens receiving T2 than in chickens receiving T3 but was not significantly different from T1. On the other hand, TFI and FCR were unaffected by all dietary treatments (*p* > 0.05).

### 3.3. Carcass Traits and Meat Quality

The effects of the addition of puncture vine on carcass characteristics and lymphoid organs in broiler chickens are presented in Table 4. All carcass characteristics and lymphoid organs were unaffected by the dietary treatments (*p* > 0.05), except for the relative leg and gizzard weights, which were higher in chickens receiving T2 (*p* = 0.003 and *p* = 0.017, respectively) than in the T3 and control (T1) groups.

The effects of the addition of puncture vine on the physical characteristics of meat quality indicators in broiler chickens are presented in Table 5. In the current study, the relative weight of cooking water loss increased (*p* = 0.047) while the myofibril fragmentation index decreased (*p* = 0.022) in chickens receiving puncture vine (T2 and T3) compared to T1. In contrast, the percentage of the water-holding capacity and texture profile parameters such as hardness, springiness, cohesiveness, and chewiness were unaffected by all dietary treatments (*p* > 0.05).

The pH value (initial and ultimate) was not influenced by any of the dietary treatments (*p* > 0.05). In contrast, the measurement of the color (redness) of the meat components after slaughter and after 24 h was lower (*p* = 0.039 and *p* = 0.021, respectively) in chickens fed T3 compared to the control diet (T1) but not significantly different than T2.

### 3.4. Blood Indicators

The effects of the addition of puncture vine on the metabolic and immunological indicators in the serum of broiler chickens are presented in Table 6. Total protein and globulin were higher in chickens receiving T2 and T3 (*p* = 0.005 and *p* = 0.002, respectively) than in chickens receiving T1, while other metabolic and immunologic indicators remained unaffected by dietary treatments at 14 days old (*p* > 0.05). On the other hand, at the end of the study (35 days old), all serum metabolic indicators were unaffected by all dietary treatments (*p* > 0.05). In contrast, interleukins (*IL-4*, *IL-6*, and *IL-10*) and tumor necrosis factor-α (*TNF-α*) were higher in chickens receiving T2 than T1 and T3 at 35 days old (*p* = 0.002, *p* = 0.016, *p* = 0.014, and *p* = 0.017, respectively).

### 3.5. Selected Cecal Microbiota Content

The effects of the addition of puncture vine on the content of selected cecal microbiota in broiler chickens are presented in Table 7. The activity content of *Lactobacillus* spp. and the ratio of *Lactobacillus* spp. to *Escherichia coli* were higher in chickens receiving T2 than in T3 and T1 (*p* = 0.001). The levels of pathogenic bacteria such as *Escherichia coli* and *Salmonella* spp. were lower in chickens receiving T2 and T3 compared to T1 (*p* = 0.001 and *p* = 0.004, respectively). There was no significant change in aerobic bacteria by all diet treatments (*p* > 0.05).

### 3.6. Cecal Short-Chain Fatty Acid

The effects of the addition of puncture vine on the content of short-chain fatty acids in the cecum of broiler chickens are presented in Table 8. In the current study, acetic acid concentration increased (*p* = 0.006) while propionic acid concentration decreased (*p* = 0.003) in chickens receiving T2 compared to T1 and T3. In contrast, chickens receiving T3 had a higher butyric acid concentration than the other dietary treatments (*p* = 0.012). Total short-chain fatty acids were higher in chickens receiving puncture vine (T2 and T3) than in T1 (*p* = 0.001).

## 4. Discussion

Puncture vine (*Tribulus terrestris*) is a medicinal plant used as a traditional human medicine in various countries (with preventive and therapeutic effects on many diseases) because it contains bioactive compounds, including saponins, glycosides, alkaloids, flavonoids, phenols, and tannins, as shown in previous studies [39,40,41]. In the current study, gas chromatography–mass spectrometry analysis showed that the major compounds of the methanolic extract of puncture vine were 5-octadecene and hexadecanoic acid, which were the most abundant residues and accounted for 21.35 and 16.35% of bioactive compounds in puncture vine, respectively. These results agree with those of Ammar et al. [42] and Khalid et al. [43], who reported that puncture vine contains many pharmacological components belonging to flavonoids and phenolics. Nevertheless, the extraction technique and the plant parts used influence the chemical composition of puncture vine [44,45].

In the current study, no significant change in growth performance was observed among all diet treatments in the starter phase, while in the finisher phase or average feeding phases (1–35 days old), BW, BWG, and PEF of broilers receiving T3 were lower and did not change significantly between T2 and the control group (T1). From these results, it appears that the addition of puncture vine (T2, 0.08% puncture vine) had no negative effect on growth performance, whereas increasing the level to 0.16% (T3) resulted in a corresponding decrease in broiler performance. However, our results agree with those of Duru and Şahin [22], who found that broiler production performance was not affected by feeding puncture vine (0.1 and 0.2%/kg basal diet). Duru and Şahin [21] reported that the inclusion of 0.006 and 0.012% puncture vine extract per kg of diet did not alter growth performance in broilers; therefore, a higher dosage of puncture vine is required. In contrast, studies by Şahin [17] and Nikolova et al. [46] found that puncture vine powder can be used in broiler diets as an alternative to antibiotics. Broilers fed with puncture vine (0.1% puncture vine) had a higher body weight compared to the control group [20]. This could be due to increased intestinal absorption due to the presence of saponins in the puncture vine [47].

Carcass characteristics were not affected by any of the dietary treatments, except that the relative weights of the legs and gizzard were higher in T2 than in T1 and T3. This suggests that the high relative weight of the legs and gizzard due to the addition of 0.08% puncture vine (T2) could be related to the final body weight before slaughter. Determining the quality of meat, especially breast meat, based on its nutritional and commercial value plays an important role as a strategic indicator for knowing the productivity of broiler chickens, the food industry, and the purchasing power of humans by focusing on nutrition and feed additives [48]. The results showed that meat quality indicators were not affected by all dietary treatments, except that the relative weight of cooking losses was higher and the myofibril fragmentation index was lower in chickens fed T2 and T3 than in the control diet. Water loss during cooking is due to shrinkage and may therefore be directly correlated with the loss of juiciness [49]. In chickens fed puncture vine (T2 and T3), cooking loss was higher, so the probability of moisture loss during cooking was higher, which could be due to the loss of juiciness due to the bioactive components of puncture vine. As a result, the myofibril fragmentation index decreased in chickens fed puncture vine compared to T1, suggesting that the puncture vine may have made the meat less firm. The color (redness) of the meat components was lower in T3 than T1 but not significantly different than T2. The addition of puncture vine and the carcasses of chickens both before and after slaughter may affect the color (redness) of the breast meat. Other factors that can affect meat color variations are heme pigments, moisture content, strain, and the physical state of the protein [50].

However, the current research results are in agreement with those of Yazdi et al. [20], who found no effect on carcass and meat characteristics of broiler chickens fed diets enriched with puncture vine extract. At the same time, the use of puncture vine extract did not affect the relative weight of the heart, abdominal fat, gizzard, and pancreas, while liver weight decreased in broilers receiving a high amount of 0.36 g of puncture vine extract per kg [14]. Other studies reported that the addition of puncture vine powder at levels of 0.2, 1, and 2 g/kg tended to decrease the breast weight of broilers without affecting other body components and carcass weight [22]. The primary and secondary lymphoid organs (thymus, bursa, and spleen) constitute the major immune organs responsible for adaptive immunity in broiler chickens [51]. However, the results of this study showed that primary and secondary lymphoid organs had no significant change by the dietary treatments.

Biochemical tests for blood indicators are performed to evaluate health and nutritional status, and these indices could help to predict the effect of a ration given to the birds [52]. The biochemical profile, such as total protein and globulin, was higher at T2 and T3, while other parameters were not affected by all diet treatments in the starter phase. In the finisher phase, the serum biochemical profile was not affected by the different feed treatments. The increase in total protein could indicate that puncture vine powder promotes protein anabolism in the body to increase immunoglobulin in broiler chickens during the starter phase. However, the results of the study showed that puncture vine powder (2 g/kg) did not cause a significant difference in blood parameters [22]. The addition of puncture vine extract (3 and 10 mg/kg diet) in hens decreased serum glucose and cholesterol levels, while it increased total protein in guinea fowls [53], mice [54], and bulls [19]. Amin et al. [55] reported that the decrease in glucose content by puncture vine could be due to the inhibition of gluconeogenesis, while the increase in total protein content could be due to the main effect of protodioscin in puncture vine. Cytokines such as the interleukins (*IL-4*, *IL-6,* and *IL-10*) and tumor necrosis factor alpha (*TNF-α*) are secreted by immune cells that play an important role in activating and regulating other cells and tissues that support the immune response [56]. These cytokines act by establishing a link between innate and adaptive responses and providing early signals to the immune system about potential dangers [57]. The results showed that chickens receiving T2 had a higher activation and regulation of the immune response at 35 days of age compared to T1 and T3 through increased levels of *IL-4*, *IL-6*, *IL-10,* and *TNF-α*. These results suggest that the production of inflammatory mediators and cytokines could be regulated by the bioactive components of puncture vine, especially at a dose of 0.08% (T2), which may be attributed to the positive effect on gut health and regulation of immune response through the production of inflammatory mediators and cytokines due to an increase in beneficial bacteria (*Lactobacillus* spp.) and the production of acetic acid. Tilwari et al. [23] reported that the administration of puncture vine extracts (1 and 2 mg/kg) enhanced the immunomodulatory activity by increasing the humoral antibody response in rats. On the other hand, puncture vine was found to have anti-inflammatory activity [45].

The content of *Lactobacillus* ssp. activity and the ratio of *Lactobacillus* to *Escherichia coli* were higher in chickens receiving T2 than in other treatments. The content of pathogenic bacteria such as *Escherichia coli* and *Salmonella* spp. were lower in T2 and T3. The primary bioactive chemicals present in the puncture vine could be the cause of the antibacterial effect against *Escherichia coli* and *Salmonella* spp. However, the current research results are in agreement with Mohammed [24], who indicated that puncture vine powder has antimicrobial activity against *Escherichia coli* and *Salmonella* spp. In another study, it was reported that puncture vine fruit extract was most active against Gram-positive and Gram-negative bacteria [25]. According to Giannenas et al. [58], the medicinal plant can have antibacterial effects, thereby increasing the activity of beneficial bacteria such as *Lactobacillus* spp., which reflects positively on the intestinal health and growth performance of broiler chickens. The current study showed that puncture vine resulted in higher concentrations of acetic acid (T2), butyric acid (T3), and total short-chain fatty acids (T2 and T3), while propionic acid was lower in chickens fed T2 than in the control group. These results may be due to the antimicrobial activity of puncture vine, as it inhibits pathogenic bacteria and allows beneficial bacteria to proliferate and produce short-chain fatty acids, which can potentially be beneficial to the health of the host [59].

## 5. Conclusions

In conclusion, the bioactive compounds identified in the puncture vine, especially phenolic and flavonoid compounds, may have an impact on the health of broiler chickens. Therefore, the addition of puncture vine (0.08%) to broiler diets has shown potential positive effects on various aspects of broiler health and performance. While puncture vine (0.08%) did not negatively affect growth performance in the starter phase, it slightly increased body weight, weight gain, and production efficiency in the finisher phase but did not differ from the control group. Regarding meat quality, the puncture vine had no significant effect on most parameters except for cooking loss and the myofibril fragmentation index, suggesting a potential impact on juiciness and being less firm. Puncture vine (0.08%) also enhanced the activation and regulation of the immune response by increasing the levels of *IL-4*, *IL-6*, *IL-10*, and *TNF-α*. In addition, puncture vine (0.08%) improved gut health by increasing Lactobacillus spp. and the ratio of Lactobacillus to Escherichia coli, while decreasing the levels of pathogenic bacteria. Moreover, puncture vine (0.08%) resulted in higher concentrations of acetic acid and total short-chain fatty acids. These results suggest that the use of 0.08% puncture vine as a maximum level in broiler diets could have positive effects on growth performance, meat quality, immune function, gut health, and short-chain fatty acid production. However, further research is needed to fully understand the long-term effects and potential side effects of this feed additive.

## Figures and Tables

**Table 1 animals-13-03708-t001:** Feed ingredients and nutrient content of basal diet control.

Feed Ingredients, g/kg	Basal Diet Phases	
	Starter (1–14 Days Old)	Grower (15–35 Days Old)
Corn	548.9	601.3
Soybean meal	387.7	319.2
Palm oil	19.6	42.1
Dicalcium phosphate	17.4	13.8
Limestone	11.9	10.5
Salt	4.0	4.0
Min + Vit Premix ^a^	5.0	5.0
DL-Methionine	3.1	2.5
L-Lysine-HCL	1.3	0.8
L-Threonine	0.7	0.2
Choline CL-70%	0.5	0.5
Total	1000	1000
**Nutrient composition, g/kg**		
M.E. Kcal/kg	3000	3200
Crude Protein	230	200
Crude Fiber	22.1	21.0
Avail. P	4.8	4.1
dLys	12.8	10.6
dMet	6.4	5.5
dTSAA	9.5	8.3
dThr	8.6	7.1
dTrp	2.7	2.3
dArg	15.0	12.9
dVal	11.3	9.9

^a^ Contains per kg of premix: vitamins such as A = 2,400,000 IU, D = 1,000,000 IU, E = 16,000 IU, K = 800 mg, B1 = 600 mg, B2 = 1600 mg, B3 = 8000 mg, B5 = 3000 mg, B6 = 1000 mg, B7 = 40 mg, B9 = 400 mg, and B12 = 6 mg and minerals like Cu = 2000 mg, Fe = 1200 mg, Mn = 18,000 mg, Se = 60 mg, Co = 80 mg, I = 400 mg, and Zn = 14,000 mg.

**Table 2 animals-13-03708-t002:** The bioactive compounds identified in the methanolic extract of puncture vine ^1^.

Retention Time (min)	Chemical Compounds	Quality, %
3.90	1H-Indene, 1-ethylideneoctahydro-7a-methyl	4.24
4.07	Benzoic acid, 2-methoxy-, methyl ester	1.59
4.15	2-Ethylacridine	2.49
4.26	5-Octadecene	21.35
4.46	Hexadecanoic acid, methyl ester	16.35
4.74	Benzenepropanoic acid ester	1.04
5.10	n-Hexadecanoic acid	1.76
6.66	3,5-di-tert-Butyl-4-hydroxyphenyl propionic acid (Fenozan)	2.88
6.74	10,13-Octadecadienoic acid, methyl ester	3.42
7.85	cis-13-Octadecenoic acid, methyl ester	2.76
7.95	Octadecanoic acid, methyl ester	6.86
8.94	Octadecadienoic acid	1.51
10.33	3-(azepan-1-yl)-1,2-benzothiazole 1,1-dioxide	1.78
11.89	1,2-Benzenedicarboxylic acid, ditridecyl ester	1.55
12.14	Gentisic acid, tri-TMS	5.72
14.29	Squalene (triterpene)	1.34
15.97	Methyl 3-(3,5-di-tert-butyl-4-hydroxyphenyl) propionate (Methylox)	5.42
19.38	2-Dodecen-1-yl(-)succinic anhydride	4.43
22.43	3, 5. beta-Cholestanol	3.42
25.18	Cholesterol	2.84

^1^ The results of chemical compounds identified using gas chromatography–mass spectroscopy.

**Table 3 animals-13-03708-t003:** Effect of addition of puncture vine on productive performance of broiler chickens during feeding phases.

Parameters ^1^	Treatments ^2^			SEM ^3^	*p*-Value
	T1	T2	T3		
**Starter phase (1–14 days old)**					
BW at 1 day, g	42.287	42.285	42.278	0.010	0.829
BW at 14 days, g	396.59	394.93	383.23	6.059	0.258
BWG, g	354.31	352.64	340.9	6.060	0.258
TFI, g	435.61	434.00	428.42	6.605	0.725
FCR, g/g	1.23	1.23	1.26	0.011	0.130
PEF	230.45	229.48	217.92	4.911	0.158
**Finisher phase (15–35 days old)**					
BW at 35 days, g	1756.31 ^a^	1759.04 ^a^	1672.69 ^b^	14.12	0.0003
BWG, g	1359.71 ^a^	1364.11 ^a^	1289.46 ^b^	14.61	0.002
TFI, g	2000.35	1916.63	1894.89	31.17	0.061
FCR, g/g	1.47	1.41	1.47	0.023	0.110
PEF	353.23 ^ab^	367.51 ^a^	336.00 ^b^	5.926	0.004

^a,b^ Means values within a row for each variable, with clarification of the significant difference in the form of superscripts (*p* < 0.05). ^1^ BW = body weight; BWG = bodyweight gain; TFI = total feed intake; FCR = feed conversion ratio; and PEF = the production efficiency factor. ^2^ Dietary treatments: T1 = birds fed the control basal diet (0.0% puncture vine), T2 = basal diet with 0.08% puncture vine, and T3 = basal diet with 0.16% puncture vine. ^3^ SEM = Standard error of means for treatment effect.

**Table 4 animals-13-03708-t004:** Effect of addition of puncture vine on the carcass traits and lymphoid organs of broiler chickens.

Parameters (%)	Treatments ^1^			SEM ^2^	*p*-Value
	T1	T2	T3		
Dressing yield	65.04	63.47	65.12	0.838	0.306
Breast	42.42	37.28	40.46	0.963	0.121
Legs	40.05 ^b^	41.37 ^a^	39.47 ^b^	0.650	0.003
Gizzard	4.17 ^b^	4.60 ^a^	3.71 ^c^	0.208	0.017
Liver	2.12	2.21	2.11	0.085	0.649
Heart	0.487	0.474	0.482	0.025	0.933
Pancreas	0.248	0.298	0.278	0.015	0.073
Abdominal fat	1.91	1.78	1.80	0.208	0.915
Thymus	0.437	0.325	0.336	0.056	0.414
Bursa	0.138	0.172	0.172	0.012	0.114
Spleen	0.158	0.135	0.140	0.014	0.407

^a–c^ Means values within a row for each variable with clarification of the significant difference in the form of superscripts (*p* < 0.05). ^1^ Dietary treatments: T1 = birds fed the control basal diet (0.0% puncture vine), T2 = basal diet with 0.08% puncture vine, and T3 = basal diet with 0.16% puncture vine. ^2^ SEM = Standard error of means for treatments effect.

**Table 5 animals-13-03708-t005:** Effect of addition of puncture vine on the meat quality indicators and physical properties of broiler chickens.

Parameters	Treatments ^1^			SEM ^2^	*p*-Value
	T1	T2	T3		
Cooking loss (%)	25.40 ^b^	29.53 ^a^	30.02 ^a^	1.106	0.047
Water-holding capacity (%)	30.32	34.37	35.03	2.086	0.298
Myofibril fragmentation index	108.60 ^a^	68.80 ^b^	62.28 ^b^	9.095	0.022
Shear force (N)	7.19	6.81	7.82	0.686	0.605
Texture Profile Analysis (TPA)					
Hardness (N)	10.89	12.40	10.29	1.523	0.623
Springiness index	0.65	0.64	0.66	0.029	0.905
Cohesiveness index	0.44	0.43	0.40	0.025	0.568
Chewiness index	2.88	3.24	2.73	0.431	0.707
Physical Properties					
Initial pH	6.44	6.36	6.42	0.044	0.447
Ultimate pH	5.81	5.83	5.80	0.017	0.556
Initial Color Components ^3^					
Lightness	41.01	42.04	43.09	0.799	0.220
Redness	6.99 ^a^	5.22 ^ab^	4.96 ^b^	0.544	0.039
Yellowness	6.94	6.54	7.15	0.412	0.579
Ultimate Color Components ^3^					
Lightness	43.41	44.68	45.63	0.766	0.157
Redness	8.19 ^a^	7.21 ^ab^	6.31 ^b^	0.414	0.021
Yellowness	11.21	12.21	11.95	0.320	0.107

^a,b^ Means values within a row for each variable with clarification of the significant difference in the form of superscripts (*p* < 0.05). ^1^ Dietary treatments: T1 = birds fed the control basal diet (0.0% puncture vine), T2 = basal diet with 0.08% puncture vine, and T3 = basal diet with 0.16% puncture vine. ^2^ SEM = Standard error of means for treatment effect. ^3^ Initial, measured at 15 min after slaughter; ultimate, measured at 24 h after slaughter.

**Table 6 animals-13-03708-t006:** Effect of addition of puncture vine on serum metabolic and immunological indicators of broiler chickens.

Parameters	Treatments ^1^			SEM ^2^	*p*-Value
	T1	T2	T3		
**At 14 days old**					
Total protein (g/dL)	2.60 ^b^	4.21 ^a^	3.74 ^a^	0.248	0.005
Albumin (g/dL)	1.39	1.61	1.48	0.088	0.228
Globulin (g/dL)	1.21 ^b^	2.60 ^a^	2.26 ^a^	0.253	0.002
Glucose (mg/dL)	237.32	199.29	225.66	12.743	0.121
Total cholesterol (mg/dL)	105.89	113.85	113.74	6.043	0.573
*IL-4* (pg/mL)	51.27	49.11	51.48	1.966	0.650
*IL-6* (pg/mL)	81.79	76.44	84.93	2.625	0.109
*IL-10* (pg/mL)	89.40	90.00	89.66	1.638	0.967
*TNF-α* (pg/mL)	35.99	37.36	34.22	1.114	0.178
**At 35 days old**					
Total protein (g/dL)	4.98	5.41	5.19	0.347	0.680
Albumin (g/dL)	2.16	1.95	2.21	0.133	0.374
Globulin (g/dL)	2.81	3.45	2.98	0.401	0.512
Glucose (mg/dL)	128.62 ^a^	114.23 ^b^	129.66 ^a^	1.933	0.001
Total cholesterol (mg/dL)	125.95	121.72	124.46	4.213	0.773
*IL-4* (pg/mL)	51.70 ^b^	57.29 ^a^	49.28 ^b^	1.220	0.002
*IL-6* (pg/mL)	65.95 ^b^	82.06 ^a^	73.57 ^b^	3.323	0.016
*IL-10* (pg/mL)	74.23 ^b^	93.07 ^a^	81.04 ^b^	3.841	0.014
*TNF-α* (pg/mL)	37.02 ^b^	41.44 ^a^	35.17 ^b^	1.328	0.017

^a,b^ Means values within a row for each variable with clarification of the significant difference in the form of superscripts (*p* < 0.05). *IL* = interleukin and *TNF-α* = tumor necrosis factor-α. ^1^ Dietary treatments: T1 = birds fed the control basal diet (0.0% puncture vine), T2 = basal diet with 0.08% puncture vine, and T3 = basal diet with 0.16% puncture vine. ^2^ SEM = Standard error of means for treatment effect.

**Table 7 animals-13-03708-t007:** Effect of addition of puncture vine on selected cecal microbiota (log^10^ colony-forming units/g digesta) in broiler chickens.

Parameters	Treatments ^1^			SEM ^2^	*p*-Value
	T1	T2	T3		
*Lactobacillus* spp.	7.18 ^b^	9.01 ^a^	7.12 ^b^	0.208	0.001
*Aerobic*	9.79	10.01	9.89	0.126	0.494
*Escherichia coli*	4.87 ^a^	4.66 ^c^	4.82 ^b^	0.011	0.001
*Salmonella* spp.	4.06 ^a^	3.74 ^b^	3.72 ^b^	0.064	0.004
*Lactobacillus*/*Escherichia coli*	1.47 ^b^	1.93 ^a^	1.47 ^b^	0.044	0.001

^a–c^ Means values within a row for each variable with clarification of the significant difference in the form of superscripts (*p* < 0.05). ^1^ Dietary treatments: T1 = birds fed the control basal diet (0.0% puncture vine), T2 = basal diet with 0.08% puncture vine, and T3 = basal diet with 0.16% puncture vine. ^2^ SEM = standard error of means for treatment effect.

**Table 8 animals-13-03708-t008:** Effect of addition of puncture vine on cecal short-chain fatty acid in broiler chickens.

Parameters	Treatments ^1^			SEM ^2^	*p*-Value
	T1	T2	T3		
Acetic acid (%)	85.67 ^b^	89.73 ^a^	84.52 ^b^	0.894	0.006
Propionic acid (%)	9.40 ^a^	4.90 ^b^	8.95 ^a^	0.725	0.003
Butyric acid (%)	4.92 ^b^	5.36 ^b^	6.53 ^a^	0.306	0.012
Total short-chain fatty acid (mg/g)	45.76 ^b^	73.08 ^a^	78.87 ^a^	4.736	0.001

^a,b^ Means values within a row for each variable with clarification of the significant difference in the form of superscripts (*p* < 0.05). ^1^ Dietary treatments: T1 = birds fed the control basal diet (0.0% puncture vine), T2 = basal diet with 0.08% puncture vine, and T3 = basal diet with 0.16% puncture vine. ^2^ SEM = standard error of means for treatment effect.

## Data Availability

The data and analyses presented in this paper are freely available from the corresponding author (H.H. Al-Baadani).

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
