# Peer review of "Effect of Puncture Vine on Growth Performance, Carcass and Meat Traits, Metabolic and Immunological Blood Indicators, and Selected Cecal Microbiota in Broiler Chickens"

_animals, 2023, doi:10.3390/ani13233708_

Round 1
Reviewer 1 Report
Comments and Suggestions for Authors
The manuscript sought to evaluate a series of metabolic and immunological variables, along with zootechnical performance. The text needs revision.
a) the term "supplementation" is used incorrectly throughout the text, as supplementing is adding something that is already in the basal diet; which is not the case here. Review.
b) in the introduction section it was not clear what gap in knowledge exists that would justify this research. Make that clear.
c) It was not clear why these cytokines were measured.
d) The authors did not evaluate the microbiota, but rather qualified some specific bacteria. I made this clearer throughout the text.
e) Justification for the doses tested? because you didn't do 4 treatments, that way you could have calculated the ideal dose in a regression analysis.
f) the conclusion needs to be revised; it must respond to the objective.
Author Response
Date: 13 November, 2023
Dear Reviewer 1,
On behalf of my co-authors, I would like to express our sincere gratitude for your consideration of our manuscript titled "Effect of Puncture Vine on Carcass and Meat Traits, Metabolic and Immunological Blood Indicators, and Cecal Microbiota in Broiler Chickens." We have meticulously addressed each of the reviewers' insightful comments and incorporated their suggestions into the revised manuscript. The modifications have been highlighted using the "Track Changes" feature, allowing for easy identification of the revisions made. Additionally, we have engaged a professional linguistic reviewer to ensure the clarity and accuracy of the manuscript's language.
We are confident that the revised manuscript addresses the reviewers' concerns and presents our research findings in a more comprehensive and impactful manner. We eagerly await your further evaluation and hope that the manuscript meets the high standards of your publication.
Thank you for your time and consideration.
Sincerely,
Best regards!
Dr. Hani H. Al-Baadani    
Q1: The term "supplementation" is used incorrectly throughout the text, as supplementing is adding something that is already in the basal diet; which is not the case here. Review.
Authors’ Response: Thank you very much for this suggestion. Done as requested in all manuscript.
Q2: In the introduction section it was not clear what gap in knowledge exists that would justify this research. Make that clear.
Authors’ Response: Thank you for pointing out that. It is correct. We have revised in lines 81-89.
Q3: It was not clear why these cytokines were measured.
Authors’ Response: Thanks so much for your notification on this point. It has been modified correctly in lines 413-425.
Q4: The authors did not evaluate the microbiota, but rather qualified some specific bacteria. I made this clearer throughout the text.
Authors’ Response: It is correct. Done as requested in all manuscript.
Q5: Justification for the doses tested? because you didn't do 4 treatments, that way you could have calculated the ideal dose in a regression analysis.
Authors’ Response:
- Puncture vine (Tribulus Terrestris) is a medicinal plant accepted and recognized by humans as a traditional remedy (whole plant or fruit) for many diseases and the pharmaceutical industry. Additionally, many studies have used Puncture vine fruit as a feed additive to improve the health and reproductive capacity of roosters, rats, and humans. But we are interested in poultry. We relied on previous studies which were conducted on chickens and mice that the addition of Puncture vine to diets up to a dose of 1 g/kg is safe by improving or not showing any effect on performance, but some studies indicated that an overdose of Puncture vine (1.5 g/kg) had a negative impact on performance, with the mechanisms of impact not being determined and no further investigations taken (as indicated in our study).
- Although studies show some effects of Puncture vine, there is still a lot of controversy about the possible action mechanisms, the function, and the appropriate dose of Puncture vine. As well, there are a limited number of studies in which Puncture vine powder as a whole plant is added to the diet on the performance and general health in the normal status of broilers. All the previous justifications are hypotheses for our study.
- This indicates that Puncture vine must be studied in the future to determine the ideal dose, all authors in this study are currently working on designing future studies on the use of different levels of doses less than 1.5 g/kg diet as feed additives in broiler chickens.
We hope that all of these above points will be satisfying to you due to some of your concerns regarding the study described in the manuscript
Q6: The conclusion needs to be revised; it must respond to the objective.
Authors’ Response: It has been modified from our point of view. We, as authors, thank you for your valuable comments and questions, which gave me the opportunity to improve the manuscript.
The manuscript has been completely revised, with some language changes made and improved from our point of view. On the other hand, I thank you for your valuable comments and questions, which gave me the opportunity to improve the throughout manuscript.
Please if there are any opinions, guide us to correct it.
satisfactory for you
Thank you very much for your efforts. Your feedback is very valuable and will improve my research skills and biological knowledge for my future studies.
--
Dr. Hani H. Al-Baadani
Reviewer 2 Report
Comments and Suggestions for Authors
"Effect of Puncture Vine (Tribulus terrestris) on Carcass and Meat Traits, Metabolic and Immunological Blood Indicators, and Cecal Microbiota in Broiler Chickens"
The introduction section should start by providing background information on the importance of poultry production for the global food industry and emphasize the role of broiler chickens in meeting the increasing demand for poultry meat. It should then introduce the concept of using feed additives to enhance broiler performance, health, and carcass traits. Additionally, the rationale for investigating the impact of Puncture Vine (Tribulus terrestris) as a potential feed additive should be stated, including any existing evidence suggesting its potential benefits for broiler chickens. Finally, clearly state the objective of the study.
In the methods section, provide detailed information about the statistical methods used in the study. Specify the specific statistical tests employed, including any software or tools used for analysis. If any assumptions were made, state them explicitly. Also, mention the significance level (alpha value) used for hypothesis testing. If applicable, describe any post-hoc tests or corrections for multiple comparisons.
Revise the title to include "Growth Performance" for comprehensive coverage. Similarly, in the conclusion section, elaborate on the observed effects of Puncture Vine on broiler growth performance, providing specific data or trends. Additionally, highlight any significant findings related to carcass and meat traits, metabolic and immunological blood indicators, and cecal microbiota. Summarize the practical implications of the study's results for the poultry industry.
Author Response
Date: 13 November, 2023
Dear Reviewer 2,
On behalf of my co-authors, I would like to express our sincere gratitude for your consideration of our manuscript titled "Effect of Puncture Vine on Carcass and Meat Traits, Metabolic and Immunological Blood Indicators, and Cecal Microbiota in Broiler Chickens." We have meticulously addressed each of the reviewers' insightful comments and incorporated their suggestions into the revised manuscript. The modifications have been highlighted using the "Track Changes" feature, allowing for easy identification of the revisions made. Additionally, we have engaged a professional linguistic reviewer to ensure the clarity and accuracy of the manuscript's language.
We are confident that the revised manuscript addresses the reviewers' concerns and presents our research findings in a more comprehensive and impactful manner. We eagerly await your further evaluation and hope that the manuscript meets the high standards of your publication.
Thank you for your time and consideration.
Sincerely,
Best regards!
Dr. Hani H. Al-Baadani
Q1: The introduction section should start by providing background information on the importance of poultry production for the global food industry and emphasize the role of broiler chickens in meeting the increasing demand for poultry meat. It should then introduce the concept of using feed additives to enhance broiler performance, health, and carcass traits. Additionally, the rationale for investigating the impact of Puncture Vine (Tribulus terrestris) as a potential feed additive should be stated, including any existing evidence suggesting its potential benefits for broiler chickens. Finally, clearly state the objective of the study.
Authors’ Response: Thanks for your feedback. Done as requested in introduction (lines 40 to 46).
Q2: In the methods section, provide detailed information about the statistical methods used in the study. Specify the specific statistical tests employed, including any software or tools used for analysis. If any assumptions were made, state them explicitly. Also, mention the significance level (alpha value) used for hypothesis testing. If applicable, describe any post-hoc tests or corrections for multiple comparisons.
Authors’ Response: Thank you for pointing out that. Done as requested in methods section (lines 206-218).
Q3: Revise the title to include "Growth Performance" for comprehensive coverage.
Authors’ Response: Thanks for your feedback. Done as requested in title.
Q4: In the conclusion section, elaborate on the observed effects of Puncture Vine on broiler growth performance, providing specific data or trends. Additionally, highlight any significant findings related to carcass and meat traits, metabolic and immunological blood indicators, and cecal microbiota. Summarize the practical implications of the study's results for the poultry industry.
Authors’ Response: Thank you for this important point. We have therefore highlighted in the conclusion important results related to carcass and meat characteristics, metabolic and immune blood indicators and cecal microbiota, which clarifies the conclusion of the current study.
The manuscript has been completely revised, with some language changes made and improved from our point of view. On the other hand, I thank you for your valuable comments and questions, which gave me the opportunity to improve the throughout manuscript.
Please if there are any opinions, guide us to correct it.
satisfactory for you
Thank you very much for your efforts. Your feedback is very valuable and will improve my research skills and biological knowledge for my future studies.
Dr. Hani H. Al-Baadani
Reviewer 3 Report
Comments and Suggestions for Authors
Overall this is an interesting paper and it is clear a lot of work went into the research. However, there are some concerns about the manuscript. Broilers are normally raised on the floor with litter; however this research was conducted in battery cages. It will be difficult to transfer these results to commercial broiler production since the rearing conditions are not the same. Please explain why a CRD was used and a RCB design.
The discussion section of this manuscript needs more detail and explanation. Details as to why the authors think the legs and gizzards were heavier with the T2 treatment. A more detailed discussion about the meat quality parameters is also needed. Why do the authors think the redness and myofibril fragment index was impacted? What is the significance of these measurements as it relates to meat quality? What is preferred? In regards to the total protein and globulin results, was this expected? What values are beneficial, higher or lower values? Why would the treatments impact these levels? The authors also state that interleukins may regulated by the compounds found in the puncture vine, however the higher levels were not found with the T3 treatment. Why do you think that is so?
Line 190: Table 2, an explanation is needed for RT
Lines 197-199: “However, in the grower phase (15-35)…(T1),” Suggest revising this statement as the PEF was not significant between T1 and T3.
Line 235-236: Table 5. The treatments are labeled as C, T1, and T2 in the table, however throughout the manuscript the treatments are refereed to T1, T2, and T3. Suggest revising.
Table 5: suggest including an explanation of the initial color components in the table.
Comments on the Quality of English LanguageMinor corrections needed.
Author Response
Date: 13 November, 2023
Dear Reviewer 3,
On behalf of my co-authors, I would like to express our sincere gratitude for your consideration of our manuscript titled "Effect of Puncture Vine on Carcass and Meat Traits, Metabolic and Immunological Blood Indicators, and Cecal Microbiota in Broiler Chickens." We have meticulously addressed each of the reviewers' insightful comments and incorporated their suggestions into the revised manuscript. The modifications have been highlighted using the "Track Changes" feature, allowing for easy identification of the revisions made. Additionally, we have engaged a professional linguistic reviewer to ensure the clarity and accuracy of the manuscript's language.
We are confident that the revised manuscript addresses the reviewers' concerns and presents our research findings in a more comprehensive and impactful manner. We eagerly await your further evaluation and hope that the manuscript meets the high standards of your publication.
Thank you for your time and consideration.
Sincerely,
Best regards!
Dr. Hani H. Al-Baadani
Q1: Overall this is an interesting paper and it is clear a lot of work went into the research.
Authors’ Response: Thanks so much, I appreciate your efforts in your valuable comments and question, which gave me the opportunity to improve manuscript.
Please if there are any opinions, guide us to correct it.
Q2: Broilers are normally raised on the floor with litter; however this research was conducted in battery cages. It will be difficult to transfer these results to commercial broiler production since the rearing conditions are not the same.
Authors’ Response: Majority of broiler production in KSA are raised in cages.
On the other hand, modern broiler chickens are reared in cages, with many studies suggesting that performance is close. Also, many experiments in the literature review have been conducted in cages.
Q3: Please explain why a CRD was used and a RCB design.
Authors’ Response: Thanks so much for your notification. A block design was used because the housing system in which the study was conducted may differ between cage levels. However, the use of blocks was justified inside study environment was justified to exclude any effect on the parameters studied.
Q4: The discussion section of this manuscript needs more detail and explanation.
Authors’ Response: This has been done, as requested in this section, in all the commentaries you mentioned and from our point of view.
Q5: Details as to why the authors think the legs and gizzards were heavier with the T2 treatment.
Authors’ Response: Thanks for your feedback. Done as requested in line 364 and 365.
Q6: A more detailed discussion about the meat quality parameters is also needed.
Authors’ Response: Thanks so much for your notification on this point. It has been modified correctly in lines 366-388.
Q7: Why do the authors think the redness and myofibril fragment index was impacted? What is the significance of these measurements as it relates to meat quality? What is preferred?
Authors’ Response: We have been modified in lines 374-378 and 382-385.
Q8: In regards to the total protein and globulin results, was this expected? What values are beneficial, higher or lower values? Why would the treatments impact these levels?
Authors’ Response: Done as requested in lines 403-405.
Q9: The authors also state that interleukins may regulated by the compounds found in the puncture vine, however the higher levels were not found with the T3 treatment. Why do you think that is so?
Authors’ Response: Thanks for your feedback. Done as requested in line 418-425.
Q10: Line 190: Table 2, an explanation is needed for RT
Authors’ Response: The meaning of TR has been added in Table 2.
Q11: Lines 197-199: “However, in the grower phase (15-35)…(T1),” Suggest revising this statement as the PEF was not significant between T1 and T3.
Authors’ Response: Thanks so much for your notification. Done as requested.
Q12: Line 235-236: Table 5. The treatments are labeled as C, T1, and T2 in the table, however throughout the manuscript the treatments are refereed to T1, T2, and T3. Suggest revising.
Authors’ Response: There was an error and it has since been corrected in Table 5.
Q13: Table 5: suggest including an explanation of the initial color components in the table.
Authors’ Response: Thanks so much for your notification on this point. Done as requested in Table 5.
Q14: Minor corrections needed.
Authors’ Response: All comments have been made point by point, as required in all texts of the manuscript.
The manuscript has been completely revised, with some language changes made and improved from our point of view. On the other hand, I thank you for your valuable comments and questions, which gave me the opportunity to improve the throughout manuscript.
Please if there are any opinions, guide us to correct it.
satisfactory for you
Thank you very much for your efforts. Your feedback is very valuable and will improve my research skills and biological knowledge for my future studies.
--
Dr. Hani H. Al-Baadani
Round 2
Reviewer 1 Report
Comments and Suggestions for Authors
adjusted were done. Accept